# Early Assessment of Chemoradiotherapy Response for Locally Advanced Pancreatic Ductal Adenocarcinoma by Dynamic Contrast-Enhanced Ultrasound

**DOI:** 10.3390/diagnostics12112662

**Published:** 2022-11-02

**Authors:** Xiu-Yun Lu, Xi Guo, Qi Zhang, Yi-Jie Qiu, Dan Zuo, Sheng Chen, Xiao-Fan Tian, Yu-Hong Zhou, Yi Dong, Wen-Ping Wang

**Affiliations:** 1Shanghai Institute of Medical Imaging, Shanghai 200032, China; 2Department of Ultrasound, Zhongshan Hospital, Fudan University, Shanghai 200032, China; 3Department of Medical Oncology, Zhongshan Hospital, Fudan University, Shanghai 200032, China; 4Department of Ultrasound, Xinhua Hospital Affiliated to Shanghai Jiaotong University School of Medicine, Shanghai 200092, China

**Keywords:** dynamic contrast-enhanced ultrasound (DCE-US), chemoradiotherapy (CRT), locally advanced pancreatic ductal adenocarcinoma (LAPC), treatment response, time–intensity curves (TICs), quantitative parameters

## Abstract

Objective: To evaluate the value of dynamic contrast-enhanced ultrasound (DCE-US) and quantitative parameters in early prediction of tumor response to chemoradiotherapy (CRT) in patients with locally advanced pancreatic ductal adenocarcinoma (LAPC). Patients and Methods: In this prospective study, patients with biopsy-proved and histopathologically proved LAPC who underwent regular CRT were recruited. DCE-US evaluations were performed before and four months after CRT. SonoVue-enhanced contrast-enhanced ultrasound (CEUS) was performed by an ultrasound system (ACUSON Sequoia; Siemens Medical Solutions, USA) equipped with a 5C1 MHz convex array transducer. Time–intensity curves were created by VueBox software (Bracco, Italy), and various DCE-US quantitative parameters were obtained. Taking Response Evaluation Criteria in Solid Tumors (RECIST) based on computed tomography (CT) or magnetic resonance imaging (MRI) as the gold standard, DCE-US parameters were compared between the treatment responder group (RG) and non-responder group (NRG). The correlation between the DCE-US parameters and the serum carbohydrate antigen 19-9 (CA 19-9) level was also analyzed. Results: Finally, 21 LAPC patients (mean age 59.3 ± 7.2 years) were included. In comparing the RG (n = 18) and NRG (n = 3), no significant change could be found among the mean size of the lesions (31.2 ± 8.1 mm vs. 27.2 ± 8.3 mm, *p* = 0.135). In comparing the TICs between the two groups, the LAPC lesions in the RG took a longer time to reach peak enhancement and to wash out. Among all the DCE-US parameters, RT (rise time), WiAUC (wash-in area under the curve), WoAUC (wash-out area under the curve) and WiWoAUC (wash-in and wash-out area under the curve) decreased significantly after CRT in the RG (*p* < 0.05). The RT ratio, WiAUC ratio, WoAUC ratio and WiWoAUC ratio were closely correlated with the change in serum CA 19-9 level in the RG (*p* < 0.05). Conclusion: DCE-US might be a potential imaging method for non-invasive follow-up for early response in LAPC patients treated by CRT.

## 1. Introduction

Pancreatic ductal carcinoma (PDAC), as the fourth leading cause of carcinoma death, has a dismal prognosis with a 5-year survival rate of approximately 10% [1]. Due to its deep retroperitoneal location and untypical clinical symptoms, diagnosis of locally advanced pancreatic carcinoma (LAPC) is established in 35–40 % of patients at the time of first diagnosis [2]. LAPC is defined as surgical unresectable when it has one or more characteristics, such as extensive peripancreatic lymph node involvement, distant metastases, encasement or occlusion of the superior mesenteric vein (SMV) or SMV/portal vein confluence and involvement of the superior mesenteric artery (SMA), celiac axis, inferior vena cava or aorta [3,4]. Surgery is the only way to cure pancreatic carcinoma; there have been many native surgical techniques for it, such as endoscopic ultrasound-guided tumor ablation combined with celiac plexus neurolysis [5]. Adjuvant therapeutic protocols are recommended for LAPC patients, including chemoradiotherapy (CRT), high-intensity focused ultrasound (HIFU), proton beam therapy, etc. [6,7,8] However, adverse effects and cost-efficiency are vital concerns during the follow-up of CRT [9,10]. Early evaluation and dynamic non-invasive follow-up of the therapeutic response during CRT is essential to maximize clinical benefits and minimize adverse effects [11].

Currently, Response Evaluation Criteria in Solid Tumors (RECIST) based on computed tomography (CT) or magnetic resonance imaging (MRI) is the standard criteria for assessing the therapeutic response for CRT in pancreatic tumors [11]. It based on the change of tumor size or the appearance of necrosis changes in the tumors, the RECIST criteria is not able to evaluate the early dynamic microvascular perfusion changes after treatment [12]. Angiogenesis and microvascular perfusion changes in tumors are related to the prognosis and might appear earlier than a change in tumor size or necrosis. The current RECIST criteria may underestimate the treatment response of CRT [13,14]. Thus, dynamic imaging modalities for monitoring the therapeutic response in LAPC is urgently and clinically needed.

Dynamic contrast-enhanced ultrasound (DCE-US), as a non-invasive functional imaging modality, is recommended to evaluate the microvascular perfusion of parenchymatous organs and tumors [15,16]. Previously, it was applied to evaluate the treatment response for liver cancers, acute renal dysfunction, etc. [17,18,19] Moreover, in pancreatic disease, DCE-US showed potential value in evaluating tumor microvascular perfusion after treatments of radiotherapy and HIFU [20,21].

The purpose of this prospective study was to evaluate the value of DCE-US in the early prediction of tumor response in LAPC patients treated with CRT.

## 2. Patients and Methods

### 2.1. Institutional Board Approval

This prospective study was approved by the Institutional Review Board (No. B2020-309R). Informed consent was signed by all patients before ultrasound examination. The procedure was followed in accordance with the Declaration of Helsinki.

### 2.2. Patient Group

From January 2021 to March 2022, patients diagnosed with LAPC who underwent CRT in our institution were enrolled. All pancreatic lesions were diagnosed by endoscopic ultrasound-guided fine-needle aspiration (EUS-FNA) and histopathological results. The criteria for LAPC includes extensive peripancreatic lymph node involvement, encasement of SMV, SMA or other important vessels and distant metastases.

Inclusion criteria were: (1) patients with histologically proved pancreatic carcinoma; (2) patients with lesions diagnosed to be unresectable based on CT/MRI or PET-CT imaging results; (3) patients who planned to receive CRT in our hospital; (4) patients with DCE-US and CT/MRI examinations performed before and four months after CRT; (5) patients with complete DICOM format of ultrasound images and serum biomarkers.

Exclusion criteria were: (1) patients who received previous treatments, such as chemotherapy or radiotherapy; (2) patients with pancreatic lesions that could not be visualized clearly on B-mode ultrasound (BMUS); (3) patients without complete 2 min DICOM clips for analysis.

Patients were classified to responder group (RG) and non-responder group (NRG) according to RECIST 1.1 criteria.

### 2.3. Study Design

In this prospective study, patients with LAPC underwent DCE-US, CT/MRI imaging and serum carbohydrate antigen 19-9 (CA 19-9) level testing before and 4 months after CRT treatment. RECIST 1.1 criteria based on CT/MRI was recognized as the gold standard. DCE-US analysis was performed. TICs and quantitative parameters were compared between RG and NRG. Correlation analysis was performed between DCE-US parameters and serum CA 19-9 level (Figure 1).

### 2.4. Dynamic Contrast-Enhanced Ultrasound Examination

An ultrasound system (ACUSON Sequoia; Siemens Medical Solutions, Issaquah, WA, USA) with a 5C1 MHz convex array transducer was used in this study. First, BMUS scan was performed. The size, echogenicity and margin of pancreatic lesions were observed and recorded.

A pure blood-pool contrast agent (SonoVue^@^, Bracco, Italy) was injected as a 1.5 mL bolus into the antecubital vein, followed by a 5 mL saline flush. Patients were required to breathe steadily during CEUS examinations. The enhancement degrees and enhancement patterns were observed during the arterial phase (10–30 s), venous phase (30–120 s) and late phase (>120 s) according to current EFSUMB guidelines [22]. The cine loops of at least 2 min length in DICOM format were recorded for further analysis.

### 2.5. Quantitative Perfusion Analysis

All CEUS images were analyzed with VueBox^@^ software (Bracco, Italy) by an experienced physician with more than 5 years of pancreatic CEUS experience. Two regions of interest (ROIs) were set in the tumor as well as its surrounding pancreatic parenchyma. CEUS time–intensity curves were created and compared between the groups.

After curve fittings, results were considered credible when the quality of fit >75%. Various quantitative CEUS perfusion parameters were created and recorded, including wash-in area under the curve (WiAUC), wash-out area under the curve (WoAUC), wash-in and wash-out area under the curve (WiWoAUC), peak enhancement (PE), rise time (RT, the time from injection to the beginning of enhancement), time to peak (TTP, the period between contrast agent at the arrival in the ROI to PE), mean transit time (mTT, the period between 50% of PE), fall time (FT), wash-in perfusion index (WiPI), wash-in rate (WiR) and wash-out rate (WoR) (Table 1).

### 2.6. Treatment Response Evaluation

CT/MRI imaging was performed before and 4 months after CRT. The RECIST 1.1 was formulated to assess tumor size for clinical evaluation of therapeutic response [23]. Treatment response was defined as: complete response (CR, complete disappearance of tumor); partial response (PR, at least 30% decrease in the sum of the longest lesion’s diameters); stable disease (SD, between 30% decrease and 20% increase in sum); progressive disease (PD, more than 20% increase in sum) [24].

Then, patients were divided into a responder group (RG) and non-responder group (NRG) according to RECIST 1.1 by CT/MRI. RG included patients with CR, PR or SD response. NRG included patients with PD response.

### 2.7. Analysis between DCE-US Quantitative Parameters and Serum CA 19-9 Level

The serum CA 19-9 level of each patient was measured in the initiation and every two months during follow-up of treatment. As the clinical treatment response indicator, the serum CA 19-9 level was compared before and 4 months after CRT. Reduction in serum CA 19-9 level > 20–50% from baseline after CRT was considered clinically effective [25]. Meanwhile, the correlation between serum CA 19-9 level and DCE-US quantitative parameters was analyzed.

### 2.8. Statistical Analysis

Descriptive statistics were calculated as mean ± standard deviation for normally distributed data as well as median and interquartile ranges for the variables with non-normal distribution. Paired-sample *t* test plus Wilcoxon test were used to compare the changes in DCE-US perfusion parameters before and four months after CRT. The correlation analysis between DCE-US parameters and serum CA 19-9 level was evaluated by the Spearman’s rank correlation test.

Statistical analysis was conducted using SPSS 20.0 (IBM, Armonk, NY, USA). A *p* value < 0.05 was considered statistically significant.

## 3. Results

### 3.1. Patient Characteristics

From January 2021 to March 2022, 23 LAPC patients who met the eligibility criteria were enrolled. Among these, two cases were excluded for lack of 2 min DICOM clips. Finally, 21 patients (12 males and 9 females, mean age 59.3 ± 7.2 years) were included. No significant difference was found between the two groups in patient characteristics, including age, gender, tumor size, location or serum biomarkers (Table 2). The diagnosis of PDAC was confirmed by endoscopic ultrasound-guided fine-needle aspiration (EUS-FNA) and histopathological results. The diagnosis of LAPC was based on CT/MRI findings as invasive to surrounding vessels (n = 18, 85.7%) or peripancreatic lymph nodes (n = 4, 19.0%) or with distant metastasis (n = 8, 38.1%). Four months after CRT, patients were divided into the RG (n = 18) and NRG (n = 3) according to their radiological responses and RECIST 1.1 evaluation results (Table 2).

### 3.2. B-Mode Ultrasound (BMUS) Features

On BMUS, 10 LAPC lesions were located at the head or neck of the pancreas, 6 at the body of the pancreas and 5 at the tail of the pancreas (Table 2). All LAPC lesions showed as hypoechoic solid lesions with unclear margins before and after CRT (Figure 2). The overall size of lesions was 31.2 ± 8.1 mm vs. 27.2 ± 8.3 mm before and 4 months after CRT, respectively (*p* = 0.135).

### 3.3. Contrast-Enhanced Ultrasound (CEUS) Features

In comparing before and 4 months after CRT, all the LAPC lesions showed heterogeneous hypoenhancement during the arterial phase, venous phase and late phase of CEUS after the injection of the contrast agents (Figure 2). No significant difference could be found in the CEUS enhancement pattern or enhancement degree between the RG and NRG, nor could necrosis be found in the lesions after CRT (Figure 2).

### 3.4. Dynamic Contrast-Enhanced Ultrasound (DCE-US) Quantitative Analysis

Two ROIs were set in the tumor and surrounding parenchyma; the TIC of the pancreatic lesion as well as its surrounding pancreas parenchyma were created accordingly. When comparing the RG and NRG, it took a longer time for the pancreatic lesions in the RG to reach peak intensity and to wash out after 4 months’ treatment (Figure 3). In the RG patients, the WiWoAUC of the TIC decreased significantly from 167,651.3 (47,501.6, 340,182.0) to 158,718.6 (1261.3, 283,989.9) after 4 months’ treatment (*p* < 0.05).

Various DCE-US parameters were obtained after the curve fitting of the TICs. In comparing the DCE-US parameters between the two groups, the DCE-US quantitative parameters, including RT, WiAUC, WoAUC and WiWoAUC, decreased significantly in the RG after 4 months’ CRT (*p* < 0.05) (Table 3) (Figure 4).

### 3.5. Correlation Analysis between DCE-US Quantitative Parameters Ratio and Serum CA 19-9 Level

Changes in the DCE-US parameters and serum CA 19-9 levels in the pancreatic lesions were calculated and compared in a ratio before and 4 months after CRT. When comparing before and after 4 months’ CRT, the serum CA 19-9 level decreased in the RG patients significantly (*p* = 0.014). The changes in the DCE-US parameters, including the RT ratio, WiAUC ratio, WoAUC ratio and WiWoAUC ratio, were closely correlated with the change in serum CA 19-9 level (r = 0.5682, 0.6973, 0.5072, 0.5943, respectively; *p* < 0.05) (Figure 5).

## 4. Discussion

Adjuvant therapy might cause shrinkage in the lesions slowly, which might be found later than perfusion changes in the tumors in radiological imaging evaluations, whereas angiogenesis and microvascular perfusion changes in the tumors often occurred before morphologic change. The current RECIST criteria based on CT/MRI may underestimate the treatment response in lesions treated by adjuvant therapy [13]. Thus, it is important to assess microvascular perfusion functionally and to visualize the tumor in response to treatment in real time.

The early and accurate prediction of efficacy for treatment is of vital importance to improve the prognosis after treatment. As a cost-effective and repeatable technique, DCE-US has been proven to be able to reflect the real-time microvascular perfusion of tumors non-invasively [26,27,28]. DCE-US could evaluate in real time the microvascular perfusion of the lesions and monitor it qualitatively and quantitatively [14,15,28]. With quantitative TIC analysis, the DCE-US parameters provide a quantitative and non-invasive measurement of the microvascular perfusion changes in parenchymatous organs [15]. Previously, our study had investigated the value of DCE-US and quantitative parameters in the differentiation of pancreatic ductal adenocarcinoma and autoimmune pancreatitis with VueBox^@^ software [27]. It was also used for differential diagnosis between pancreatic carcinoma and chronic pancreatitis with quantitative contrast-enhanced harmonic EUS [29]. Moreover, quantitative endoscopic ultrasonography elastography could also accurately differentiate malignant from benign solid pancreatic masses with real-time quantification of tissue stiffness [30].

According to the current literature, DCE-US analysis could be helpful for evaluating the therapeutic effect in liver tumors treated with local ablative therapy and systemic therapy [31,32,33]. During follow-up of colorectal liver metastases receiving bevacizumab treatment, peak enhancement was proved to be a helpful parameter in predicting treatment response, with a positive predictive value and negative predictive value of 25% and 100% [34]. Previous studies used dynamic contrast-enhanced endoscopic ultrasound to differentiate pancreatic carcinoma, to predict the pathological grade and to predict treatment efficacy with quantitative methods [35,36,37]. A significant difference could be seen in relative rise intensity (rRI) between different vessel grades, which could help to assess the short-term efficacy of pulsed-wave high-intensity focused ultrasound (PW-HIFU) in advanced pancreatic cancer [38]. Currently, the RECIST criteria based on CT/MRI is regarded as the standard criteria in assessing the therapeutic response after CRT in LAPC patients. However, the role of the DCE-US analysis and quantitative parameters in evaluating the treatment response of CRT in LAPC patients has been rarely reported.

In our results, in comparing before and 4 months after CRT treatment, no significant changes could be found in the size or echogenicity of the lesions. However, there were changes both in the shape of the TICs and in the quantitative parameters, such as RT, WiAUC, WoAUC and WiWoAUC. When comparing the RG with NRG, the RG took a longer time to reach peak intensity and to wash out after treatment. In RG patients, the WiWoAUC of the TIC was found to decrease after 4 months’ treatment. Other quantitative parameters, including RT, WiAUC and WoAUC, decreased significantly in the RG. DCE-US was found to be helpful in monitoring early treatment response to CRT both in the RG and NRG groups. Compared to CT/MRI, DCE-US has unique advantages, such as non-invasiveness, real-time images and quantitative analysis.

As an effective serum biomarker for treatment response, the serum CA 19-9 level has predictive value in the long-term survival rate of patients after therapy [39]. According to the current literature, normalization of the serum CA 19-9 level or > 20–50% reduction from baseline after therapy predict longer survival [25,40]. In our study results, the correlation between the changes in the DCE-US parameters and serum CA 19-9 levels in LAPC patients after CRT was proved to be significant. Compared to baseline levels, a significant reduction in serum CA 19-9 was found in the RG patients after 4 months’ CRT. The changes in the DCE-US parameters, including the RT ratio, WiAUC ratio, WoAUC ratio and WiWoAUC ratio, were closely correlated with the change in serum CA 19-9 level, which indicated that the DCE-US parameters might be another useful method to predict treatment efficacy in LAPC patients after CRT.

This study had some limitations. As a prospective study design, the sample size was relatively small. Moreover, this study did not investigate the relationship between DCE-US parameters with overall survival and progression-free survival. In our study, there were only three non-responders in the NRG, which was relatively smaller than the RG.

In conclusion, DCE-US could be a potential imaging modality for non-invasive and quantitative evaluation of early response in LAPC patients treated with CRT. The DCE-US parameters, including RT, WiAUC, WoAUC and WiWoAUC, might be predictive for tumor response.

## Figures and Tables

**Figure 1 diagnostics-12-02662-f001:**
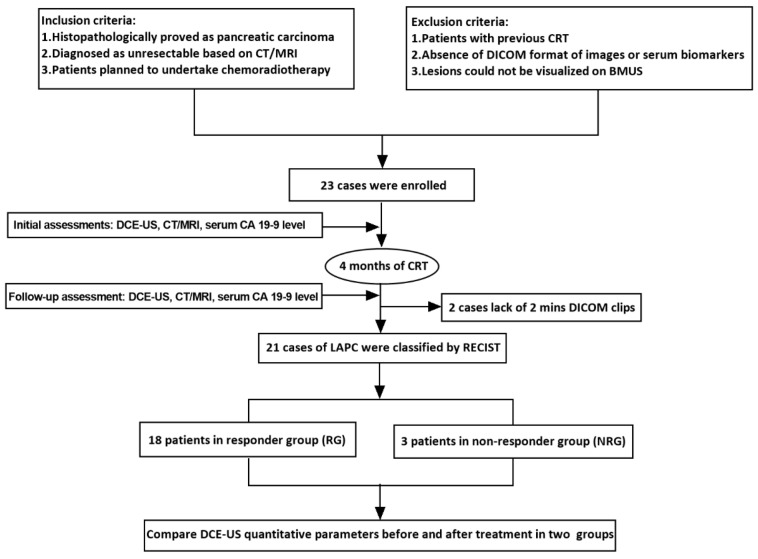
Research protocol.

**Figure 2 diagnostics-12-02662-f002:**
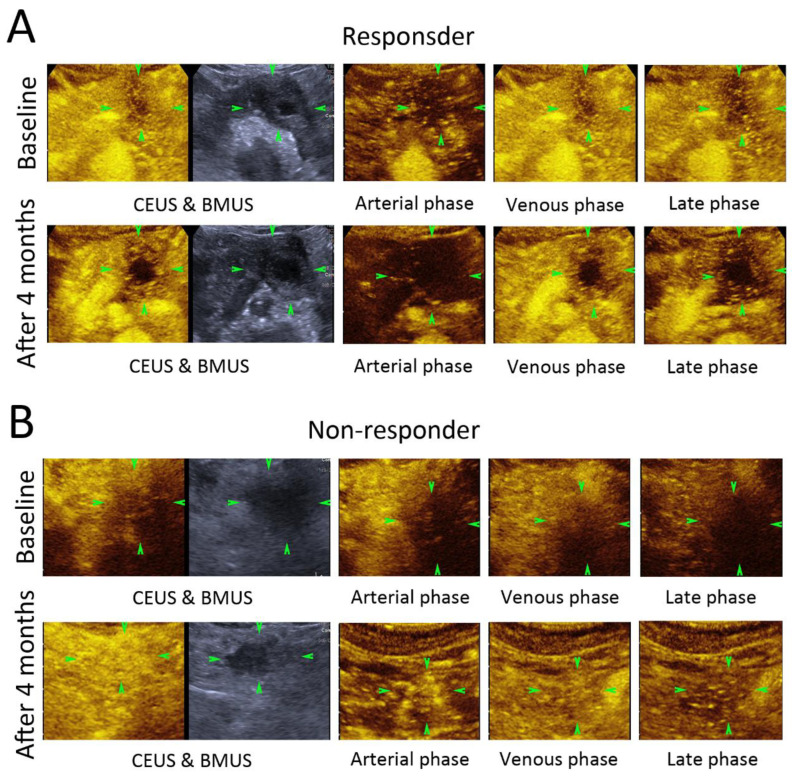
Contrast-enhanced ultrasound images of responder and non-responder before and 4 months after chemoradiotherapy (CRT). Patient as a responder to CRT (**A**). Patient as a non-responder to CRT (**B**). Comparing before and 4 months after CRT, there was no significant change in size of the tumor (green arrowheads); contrast enhancement pattern or enhancement degree could be observed either in responder or non-responder group.

**Figure 3 diagnostics-12-02662-f003:**
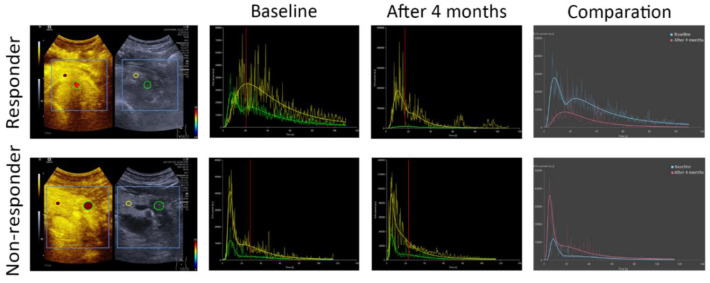
After injection of ultrasound contrast agent, time–intensity curves of locally advanced pancreatic ductal adenocarcinoma (LAPC) lesion were created. In comparing the LAPC lesion (green curve) and the surrounding pancreatic tissue (yellow curve), after chemoradiotherapy (CRT) the LAPC lesions in responder group (RG) took a longer time to reach peak enhancement and to wash out (red curve). In comparing before (blue curve) and 4 months after CRT (red curve), WiWoAUC decreased significantly more in RG than in NRG.

**Figure 4 diagnostics-12-02662-f004:**
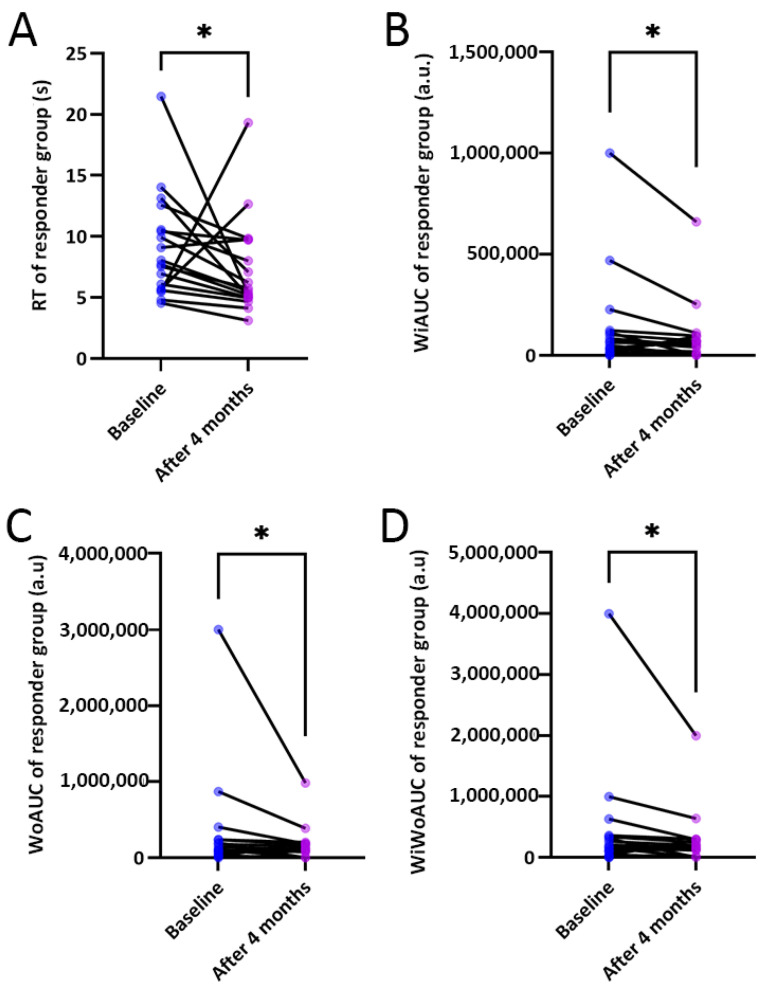
Comparison of the differences in rise time (RT), wash-in area under the curve (WiAUC), wash-out area under the curve (WoAUC) and wash-in and wash-out area under the curve (WiWoAUC) between baseline and 4 months after chemoradiotherapy (CRT) in responder group (RG) (**A**–**D**) (* *p* < 0.05).

**Figure 5 diagnostics-12-02662-f005:**
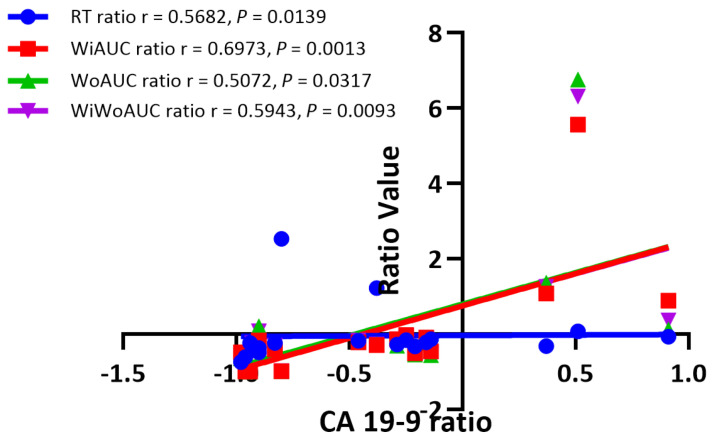
Correlation analysis between dynamic contrast-enhanced ultrasound (DCE-US) quantitative parameters and serum carbohydrate antigen 19-9 (CA 19-9) level in responder group (RG).

**Table 1 diagnostics-12-02662-t001:** Dynamic contrast-enhanced ultrasound (DCE-US) perfusion parameters.

Abbreviation	Specific Meanings
**WiAUC**	wash-in area under the curve
**WoAUC**	wash-out area under the curve
**WiWoAUC**	wash-in and wash-out area under the curve
**PE**	peak enhancement
**RT**	rise time, the time from injection to the beginning of enhancement
**TTP**	time to peak, the period between contrast agent at the arrival in the ROI to PE
**mTT**	mean transit time, the period between 50% of PE
**FT**	fall time
**WiPI**	wash-in perfusion index
**WiR**	wash-in rate
**WoR**	wash-out rate

**Table 2 diagnostics-12-02662-t002:** Patient characteristics.

Patient Characteristics	Study Population n = 21	RG n = 18	NRG n = 3	*p* Value
**Age (year)**	59.3 ± 7.2	59.8 ± 7.5	56.7 ± 4.9	0.501
**Gender**				0.553
**Male**	12 (57.1%)	11 (61.1%)	1 (33.3%)	
**Female**	9 (42.9%)	7 (38.9%)	2 (66.7%)	
**Size of tumors (mm)**				
**Baseline**	31.2 ± 8.1	30.8 ± 8.2	33.7 ± 8.6	0.587
**Four months after treatment**	27.2 ± 8.3	27.7 ± 8.7	24.0 ± 5.3	0.487
**Location of pancreatic lesions**				1.000
**Head/neck of pancreas**	10 (47.6%)	9 (50.0%)	1 (33.3%)	
**Body of pancreas**	6 (28.6%)	5 (27.8%)	1 (33.3%)	
**Tail of pancreas**	5 (23.8%)	4 (22.2%)	1 (33.3%)	
**Serum CA 19-9 level (KU/L)** **Baseline** **After 4 months**	210.0 [21.4–411.0] 25.0 [14.9–80.7]	221.5 [53.2–424.0] 31.4 [14.9–80.7]	12.0 [10.3–42.2] 19.1 [14.6–1224.1]	0.800 1.000

RG: responder group; NRG: non-responder group.

**Table 3 diagnostics-12-02662-t003:** Comparison of DCE-US parameters between responder group (RG) and non-responder group (NRG).

	Before Treatment	4 Months after CRT	*p* Value
**PE (a.u)**			
**RG**	13,176.3 (2564.4, 27,748.8)	10,202.7 (91.9, 30,433.0)	0.557
**NRG**	2592.0 (2372.8, 30,164.5)	13,851.4 (11,979.7, 36,910.8)	0.113
**WiAUC (a.u)**			
**RG**	56,215.7 (15,122.2, 112,512.4)	52,344.1 (359.1, 94,145.0)	**0.031**
**NRG**	30,839.1 (22,473.2, 138,878.9)	56,322.3 (51,738.2, 178,273.7)	**0.046**
**RT (s)**			
**RG**	7.9 (5.7, 10.5)	5.6 (5.0, 9.7)	**0.022**
**NRG**	9.4 (8.0, 14.0)	7.3 (6.9, 7.6)	0.392
**mTT (s)**			
**RG**	46.3 (29.3, 58.0)	32.0 (25.3, 46.3)	0.184
**NRG**	160.5 (100.1, 319.1)	34.6 (34.3, 43.2)	0.295
**TTP (s)**			
**RG**	13.5 (9.1, 18.3)	10.9 (7.8, 15.7)	0.263
**NRG**	10.5 (9.8, 19.2)	11.1(10.6, 11.3)	0.525
**WiR**			
**RG**	2393.2 (388.0, 6562.9)	1552.9 (20.2, 8722.2)	0.327
**NRG**	428.7 (314.8, 6796.9)	3086.7 (2535.7, 7121.1)	0.635
**WiPI**			
**RG**	8228.1 (1578.1, 17,935.0)	6309.4 (57.6, 19,455.3)	0.500
**NRG**	1654.8 (1574.3, 19,512.1)	8810.4 (7625.8, 23,578.2)	0.137
**WoAUC (a.u)**			
**RG**	108,061.2 (32,379.5, 237,872.4)	107,903.6 (956.6, 183,932.3)	**0.028**
**NRG**	59,206.9 (51,960.2, 328,627.9)	103,650.0 (100,540.2, 351,388.3)	0.178
**WiWoAUC (a.u)**			
**RG**	167,651.3 (47,501.6, 340,182.0)	158,718.6 (1261.3, 283,989.9)	**0.028**
**NRG**	90,046.0 (74,433.4, 467,506.8)	159,972.3 (152,278.4, 529,662.0)	**0.016**
**FT (s)**			
**RG**	14.1 (11.9, 23.9)	12.4 (8.4, 19.1)	0.112
**NRG**	34.8 (25.0, 36.6)	14.8 (13.6, 15.7)	0.213
**WoR**			
**RG**	1056.5 (200.2, 1967.0)	631.5 (7.6, 3172.8)	0.679
**NRG**	75.0 (70.6, 2110.7)	1236.9 (998.6, 2618.6)	0.275

PE: peak enhancement; WiAUC: wash-in area under the curve; RT: rise time; mTT: mean transit time; TTP: time to peak; WiR: wash-in rate; WiPI: wash-in perfusion index; WoAUC: wash-out area under the curve; WiWoAUC: wash-in and wash-out area under the curve; FT: fall time; WoR: wash-out rate; CRT: chemoradiotherapy.

## Data Availability

Data available on request due to ethical restrictions. The data presented in this study are available on request from the corresponding author. The data are not publicly available due to ethical restrictions.

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
