# Peer review of "Early Assessment of Chemoradiotherapy Response for Locally Advanced Pancreatic Ductal Adenocarcinoma by Dynamic Contrast-Enhanced Ultrasound"

_diagnostics, 2022, doi:10.3390/diagnostics12112662_

Round 1

Reviewer 1 Report

- Row 62-63: The rephrasing of the sentence "Thus, dynamic imaging modalities for monitoring therapeutic response in LAPC is urgently needed clinically." is recommended (e.g. "urgently and clinically needed" could be a better statement)

- Row 81-82: In this sentence "including" was not used as a verb and leaves the sentence incomplete. Thus, the revision of sentence is necessary.

- Row 83 & Figure 1: Usage of "Inclusion criteria" would be a better choice.

- Rows 83-87 & 88-90: The criteria listed in these paragraphs are not consistent in terms of subject, revision is recommended for consistency. E.g. Row 84 could be rephrased as: "2) patients with lesions that were diagnosed to be unresectable based on CT/MRI or PET-CT imaging results;".

- Row 88 & Figure 1: Usage of "Exclusion criteria" would be a better choice.

- Row 94: "... LAPC were underwent ... " were is not necessary as "underwent" is already a verb in past tense.

- Row 146: Term "not-normally distributed" should be revised. A better interpretation can be achieved by "... interquartile ranges for the variables with non-normal distribution" or "... interquartile ranges for the variables that are not normally distributed". 

- Row 223: "Shrink" should be revised to "shrinkage" to make it a noun as "to cause" is the verb of the sentence.

- Row 236: "Evaluation" should be revised as "evaluating".

- Row 240: "Nagative" should be changed to "negative". "..., with positive predictive value and nagative predictive value were 25 % and 240 %" can be revised as "..., with positive predictive and negative predictive values as 25 % and 100%, respectively" / or a similar correction needed.

- Row 91 and 93: "restrictions ethical" can be revised as "ethical restrictions".

Author Response

Dear Professor,

Thank you for the comments concerning our manuscript entitled “Early assessment of chemoradiotherapy response for locally advanced pancreatic ductal adenocarcinoma by dynamic contrast enhanced ultrasound” (Manuscript ID diagnostics-1965760). The comments have been important and helpful to improve our paper. All changes were highlighted by tracking mode in our revised manuscript. Please find our point-to-point reply below.

We are hoping that after careful revision our paper deserves publication in Diagnostics.

Thank you with Best regards!

Prof. Dr. Yi Dong

Point 1: - Row 62-63: The rephrasing of the sentence "Thus, dynamic imaging modalities for monitoring therapeutic response in LAPC is urgently needed clinically." is recommended (e.g. "urgently and clinically needed" could be a better statement)

Response 1: Thank you for your suggestion. The mistake has been corrected now.

Point 2: - Row 81-82: In this sentence "including" was not used as a verb and leaves the sentence incomplete. Thus, the revision of sentence is necessary.

Response 2: Sorry for this mistake. We have corrected according to your suggestion.

Point 3: - Row 83 & Figure 1: Usage of "Inclusion criteria" would be a better choice.

Response 3: Sorry for this mistake. We have corrected according to your suggestion.

Point 4: - Rows 83-87 & 88-90: The criteria listed in these paragraphs are not consistent in terms of subject, revision is recommended for consistency. E.g. Row 84 could be rephrased as: "2) patients with lesions that were diagnosed to be unresectable based on CT/MRI or PET-CT imaging results;".

Response 4: Thank you for your suggestion. The mistake has been corrected.

Point 5: - Row 88 & Figure 1: Usage of "Exclusion criteria" would be a better choice.

Response 5: Thank you for your suggestion. The mistake has been corrected.

Point 6: - Row 94: "... LAPC were underwent ... " were is not necessary as "underwent" is already a verb in past tense.

Response 6: Thank you for your suggestion. The mistake has been corrected. And the mistakes like this will be taken seriously.

Point 7: - Row 146: Term "not-normally distributed" should be revised. A better interpretation can be achieved by "... interquartile ranges for the variables with non-normal distribution" or "... interquartile ranges for the variables that are not normally distributed". 

Response 7: Sorry for this mistake. We have corrected according to your suggestion. More attention will be paid to the writing of nouns and adverbs in the future.

Point 8: - Row 223: "Shrink" should be revised to "shrinkage" to make it a noun as "to cause" is the verb of the sentence.

Response 8: Thank you for your suggestion. We have corrected according to your suggestion.

Point 9: - Row 236: "Evaluation" should be revised as "evaluating".

Response 9: Thank you for your suggestion. The mistake has been corrected.

Point 10: - Row 240: "Nagative" should be changed to "negative". "..., with positive predictive value and nagative predictive value were 25 % and 240 %" can be revised as "..., with positive predictive and negative predictive values as 25 % and 100%, respectively" / or a similar correction needed.

Response 10: Sorry for this mistake. We have corrected it according to your suggestion.

Point 11: - Row 91 and 93: "restrictions ethical" can be revised as "ethical restrictions".

Response 11: Thanks for your suggestion. The mistake has been corrected. And the mistakes like this will be taken seriously in the future.

Reviewer 2 Report

I really enjoyed reviewing this manuscript. However, there are some points that should be addressed by the authors:

1) I have some concerns on the validity of the comparative analysis between responders and non-responders, considering that we had only 3 non-responders.....this aspect should be adequately addressed by the authors among the limitations.

2) I would comment in the discussion also other potential roles of EUS in locally-advanced PDAC, for example citing and commenting the recent paper PMID: 27356212)

3) The authors should comment more on the potential role of VueBox in pancreatology

4) I would comment also on the potential role of EUS-elastography in this setting

Author Response

Dear Professor,

Thank you for the comments concerning our manuscript entitled “Early assessment of chemoradiotherapy response for locally advanced pancreatic ductal adenocarcinoma by dynamic contrast enhanced ultrasound” (Manuscript ID diagnostics-1965760). The comments have been important and helpful to improve our paper. All changes were highlighted by tracking mode in our revised manuscript. Please find our point-to-point reply below.

We are hoping that after careful revision our paper deserves publication in Diagnostics.

Thank you with Best regards!

Prof. Dr. Yi Dong

Point 1: I have some concerns on the validity of the comparative analysis between responders and non-responders, considering that we had only 3 non-responders.....this aspect should be adequately addressed by the authors among the limitations.

Response 1: Thank you for your suggestion. The comment was added to the discussion part with citation of these important papers.

Point 2:  I would comment in the discussion also other potential roles of EUS in locally-advanced PDAC, for example citing and commenting the recent paper PMID: 27356212)

Response 2: Thanks for your suggestion. The comment was added to the introduction and discussion part with citation of these important papers.

Point 3: The authors should comment more on the potential role of VueBox in pancreatology.

Response 3: Thanks for your suggestion. The comment was added to the discussion part with citation of important papers about the potential role of VueBox in pancreatology.

Point 4: I would comment also on the potential role of EUS-elastography in this setting.

Response 4: Thank you for your suggestion. We have added the potential role of EUS-elastography in the discussion part with citation of relative papers.

Round 2

Reviewer 2 Report

The revised version of the manuscript is OK. Thank you!